# Astrovirus infects actively secreting goblet cells and alters the gut mucus barrier

Valerie Cortez [1], David F. Boyd[2], Jeremy Chase Crawford [2], Bridgett Sharp[1], Brandi Livingston[1], Hannah M. Rowe[1], Amy Davis[1], Ramzi Alsallaq [1], Camenzind G. Robinson [3], Peter Vogel [4], Jason W. Rosch [1], Elisa Margolis[1], Paul G. Thomas [2] & Stacey Schultz-Cherry [1]✉

Astroviruses are a global cause of pediatric diarrhea, but they are largely understudied, and it is unclear how and where they replicate in the gut. Using an in vivo model, here we report that murine astrovirus preferentially infects actively secreting small intestinal goblet cells, specialized epithelial cells that maintain the mucus barrier. Consequently, virus infection alters mucus production, leading to an increase in mucus-associated bacteria and resistance to enteropathogenic *E. coli* colonization. These studies establish the main target cell type and region of the gut for productive murine astrovirus infection. They further define a mechanism by which an enteric virus can regulate the mucus barrier, induce functional changes to commensal microbial communities, and alter host susceptibility to pathogenic bacteria.

[1] Department of Infectious Diseases, St. Jude Children's Research Hospital, Memphis, TN, USA. [2] Department of Immunology, St. Jude Children's Research Hospital, Memphis, TN, USA. [3] Cell and Tissue Imaging Center, St. Jude Children's Research Hospital, Memphis, TN, USA. [4] Veterinary Pathology Core, St. Jude Children's Research Hospital, Memphis, TN, USA. ✉email: stacey.schultz-cherry@stjude.org

With an estimated 47.8 million cases worldwide, astroviruses are a leading cause of pediatric diarrhea[1,2]. But their prevalence is actually underestimated[3] due to the lack of diagnostic methods that can detect all circulating strains[2,4]. Similar to other enteric viruses, astrovirus infections can range from subclinical manifestations to severe diarrhea and even fatal encephalitis and meningitis in immunocompromised hosts[4]. However, little is known about their cellular tropism or pathogenesis, except that unlike other enteric viruses, astrovirus infections in multiple host species, including mice, are characterized by minimal cell death, inflammation, or immune cell infiltration[4,5]. Also, epithelial cells within the villi, but not the crypts, have been implicated as targets for human and turkey astroviruses[6,7], but a detailed analysis with respect to host responses to infection has not been performed.

We hypothesize that the lack of pathophysiology caused by astroviruses is related to their cellular tropism. Here we define their epithelial target in the gut and response to infection using the murine model for astrovirus[5]. We show that astrovirus specifically infects the small intestine and replicates in a subset of goblet cells that are actively secreting mucus. We further demonstrate direct consequences of infection for the host, including a simultaneous increase in the mucus barrier and mucus-associated bacteria, which translates to reduced susceptibility to enteropathogenic *E. coli* (EPEC).

## Results

**Astrovirus infects small intestinal goblet cells**. We first characterized the temporal location of murine astrovirus (MuAstV) in the small and large intestines of wild-type C57BL/6 mice (3 days post infection (dpi) to 42 dpi) using an in situ hybridization (ISH) probe set specific for the MuAstV genome. We observed staining that mirrored the replication kinetics of the virus[5], with the most abundant signal corresponding with peak infection (10 dpi) and an absence of staining once the virus was cleared (Fig. 1a) or in tissue sites that lack active replication, such as the lung (Supplementary Fig. 1a). Throughout the villi, we noted staining in cells morphologically consistent with goblet cells, a specialized epithelial cell type that produces the main components of the mucus barrier (Fig. 1a). Because staining for the goblet cell marker, mucin 2 (Muc2), was incompatible with the ISH procedures, we stained serial tissue sections and confirmed the virus' selective infection of goblet cells (Supplementary Fig. 1b). We also ruled out infection of tuft cells, which have a similar distribution as goblet cells in the intestine (Supplementary Fig. 1c). Although high virus levels can be detected by quantitative reverse transcription PCR throughout the small and large intestine[5], we only observed ISH staining within the small intestine, with the highest density of positive cells in the duodenum and jejunum (median = 30% goblet cells/villus, range = 0–90%). However, virus was clearly visualized throughout the lumen of the ileum (Supplementary Fig. 1d) and in the mucus layer of the proximal colon (Supplementary Fig. 1e), but not the distal colon (Supplementary Fig. 1f). We confirmed virus production in goblet cells using electron microscopy, which revealed virus particles clustered amongst mucin granules (Fig. 1b).

**Active goblet cell secretion is critical for replication**. Because goblet cells constitute only 4–12% of epithelial cells in the small intestine[8] and only a subset of these cells appeared to be productively infected, we next used single-cell RNA sequencing (scRNA-seq) to confirm this tropism of astrovirus and investigate the cellular response to infection. Using CD45-EpCAM+ sorted epithelial cells isolated from the duodenal villi of MuAstV-infected and mock-infected animals collected at 6 dpi, we

performed unsupervised clustering to first resolve the different epithelial cell populations based on gene profiling for cell-specific markers (Supplementary Fig. 2a–e). These epithelial populations included enterocytes, stem-like cells, enteroendocrine, tuft, and goblet cells (Fig. 2a). We analyzed a total of 2973 aggregated cells from eight animals (n = 4/infection group) and found that the cell populations largely overlapped between infected and uninfected animals (Fig. 2b) and there were no significant differences in the proportion of cells in each population ($\chi^2$ test, $p = 0.9849$), indicating that virus infection did not induce massive changes to the epithelial composition. While viral transcripts (open readings frames (ORF)1a, ORF1b, and ORF2) were detected at low levels across all epithelial clusters except for one subset of enteroendocrine cells (Supplementary Fig. 3, cluster 10) and tuft cells (Supplementary Fig. 3, cluster 12), high expression of all three viral ORFs were detected in a subset of goblet cells from infected animals (n = 57/133, 43%), which was consistent with our ISH and indicative of productive replication (Fig. 2c). Amongst all goblet cells from infected and uninfected animals, we identified four subpopulations with different gene expression profiles (Fig. 2c). However, similar to our examination of epithelial cell populations, none of these goblet cell subpopulations differed in proportion ($\chi^2$ test, $p = 0.2007$), indicating that the virus does not lead to goblet cell expansion or contraction after infection (Fig. 2c). Notably, three of the populations were susceptible to MuAstV infection (Fig. 2d). Several of the top genes that best distinguished the susceptible and nonsusceptible goblet cell clusters encode components of mucus (Supplementary Table 1, Fig. 2d), including *Muc2*, *Fcgbp*, *Clca1*, and *Zg16*, which were more highly expressed in infected animals compared with uninfected animals (Fig. 2e). Furthermore, we performed a gene set enrichment analysis comparing infected goblet cells with uninfected goblet cells and noted that pathways related to secretion, including positive regulation of secretion, vesicle-mediated transport, and exocytosis, were all significantly enriched amongst infected cells (false discovery rate (FDR) $q$ value < 0.2, Fig. 2f). While the secretory pathways have yet to be defined for small intestinal goblet cells, the enriched pathways we identified included previously described components of mucus secretion in airway and colonic goblet cells, such as syntaxins, VAMP, and SNAP proteins[9,10]. Together, these data indicate that the virus preferentially infects actively secreting goblet cells and could be driving a further increase in secretion after infection. Indeed, we noted a slightly higher proportion of secretory goblet cells in infected animals (61%, n = 81/133) compared with uninfected animals (52%, n = 46/89), but this was not statistically significant ($\chi^2$ test, $p = 0.1738$). To investigate this further we would ideally have a system to block mucus secretion; however, because mucus production is tightly regulated and there is a lack of methods that block both constitutive and induced secretion[9], we used established drug treatments that alter the gut environment and have documented detriments to goblet cell physiology and/or the mucus barrier. The first, dextran sodium sulfate (DSS), is widely used to model colitis and has been shown to cause goblet cell depletion that precedes inflammation[11], with detrimental effects have also been noted in the small intestine[12]. As a second approach, broad-spectrum antibiotics were used to drastically alter the gut microbiome and halt homeostatic cues from the microbiota to induce mucus secretion[13,14]. Pretreatment with either drug led to altered goblet cell morphology and reduced secretion based on Muc2 staining (Supplementary Fig. 4a), which significantly reduced virus infection and shedding based on qRT-PCR (Fig. 3a, b) and ISH staining (Supplementary Fig. 4b). Upon removal of DSS treatment, virus shedding returned to the same level as untreated animals (Fig. 3a). Overall, these data support that the virus targets actively secreting goblet cells for replication.

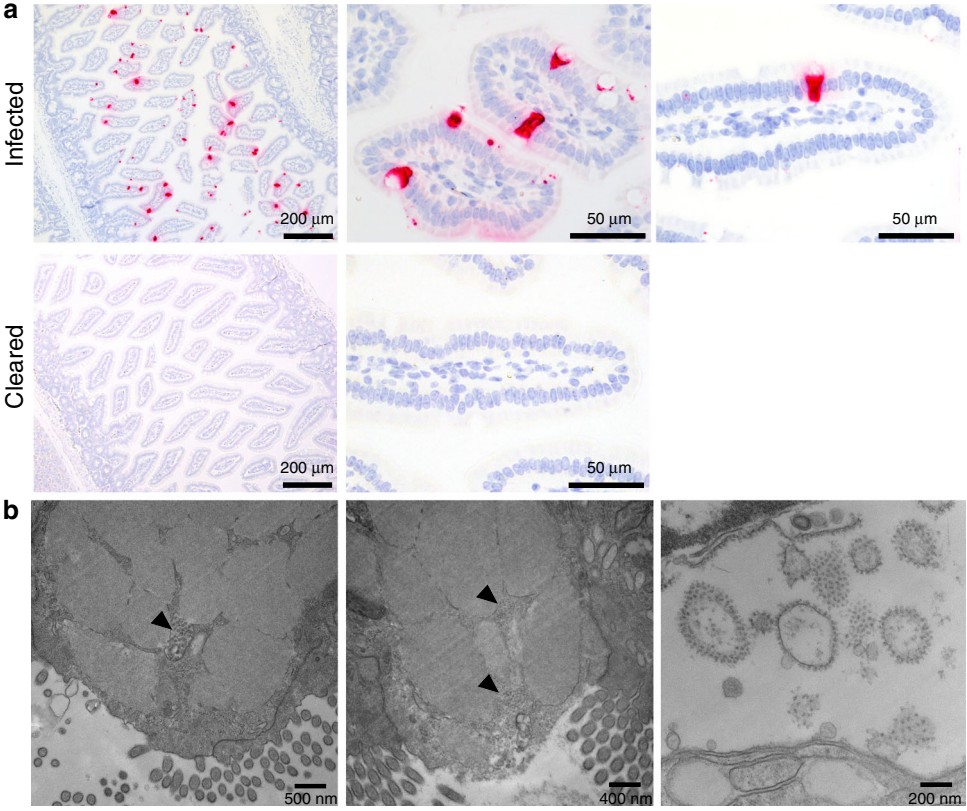

**Fig. 1 Murine astrovirus replicates in goblet cells of the small intestine. a** Small and large intestines were collected throughout infection of 8-week-old C57BL/6 mice and hybridized with murine astrovirus-specific probes shown in red. The representative images show staining of cells with distinctive goblet cell morphology in the small intestine at 10 dpi (n = 4 animals/group). **b** Duodenal goblet cells from infected animals (n = 2) harvested at 8 and 10 dpi were analyzed by electron microscopy. Virus particles were visualized amongst mucin granules (black arrowheads) and replication bodies were evident in the perinuclear space (third panel). Data shown are from two independent experiments.

However, as an orthogonal approach to these methods that disrupt secretion, we attempted to use the type II cytokines, interleukin (IL)-4 and IL-13, to induce goblet cell hyperplasia and mucus secretion[9,15,16]. However, when either treatment was delivered before (−3 and −1 dpi) and after infection (1, 3, and 5 dpi), this led to reduced virus in the tissue and feces in comparison with untreated animals (Supplementary Fig. 5a). In fact, both treatments failed to induce hallmarks of secretory hyperplasia (Supplementary Fig. 5b), which contrasts previous findings in the colon after *Citrobacter rodentium* infection[15]. Similarly, treatment of infected animals with IL-4 after at 21 dpi, when virus levels plateau, failed to induce higher levels of virus shed in the feces (Supplementary Fig. 5c), which is contrary to previous observations of tuft cell hyperplasia after IL-4 treatment leading to increased murine norovirus production[17]. Together, these data likely signify distinct regional differences in type II immune responses in the gut and highlight more complex interactions between secretory cells, mucus production, and enteric viruses than was previously appreciated. Future studies that unravel these interactions are greatly needed.

**Functional changes to the mucus barrier after infection**. Having found alterations in the transcriptional pathways involved with mucus secretion that may be enhanced by infection, we next examined whether there was a functional change to the mucus barrier. We first stained tissue sections using periodic acid-Schiff and measured mucus thickness. Because the mucus layer is more variable in the small intestine in comparison with the large intestine[18], we measured both the layer at the top and in between

villi and found that both measures were on average 1.85 to 2.51-fold higher after infection (7 dpi) in comparison with control animals (Fig. 4a). An increase in mucus can translate to an enriched carbon source for gut microbiota[19–21], so we next hypothesized that MuAstV would shift the microbiome composition towards mucus-associated bacteria. Using 16S metagenomic sequencing and analysis, we characterized the microbial communities within feces sample collected at 0, 7, and 14 dpi (Supplementary Fig. 5a). While the overall bacterial diversity did not change after infection (Supplementary Fig. 5b), when we indiscriminately examined the dataset for trends over time, we found that only the relative frequency of well-characterized mucus-associated bacteria[22,23] (Supplementary Table 2) significantly increased with time since infection (Fig. 4b). Using principal components (PCs) analysis, we also noted an increase over time in components whose phylogenetic edges represent putative mucin degraders and bile acid resistant bacteria[22–24] (Supplementary Fig. 6, Fig. 4c). To determine whether these changes to the mucus barrier and microbiome composition could alter host susceptibility to an adherent bacterial pathogen in the small intestine, we next tested whether EPEC could colonize mice after MuAstV infection. Adult animals are not colonized with EPEC efficiently without antibiotic pretreatment[25], but since we observed that antibiotic treatment can affect MuAstV replication (Fig. 3b), we developed a neonatal model using 7-day-old pups. MuAstV tropism and enhanced mucus thickness mirrored what we observed in adult mice (Supplementary Fig. 7) and coinfected pups developed significantly lower EPEC loads in comparison with pups given EPEC alone (Fig. 4d). These results are in line with a recent report demonstrating that increased mucus

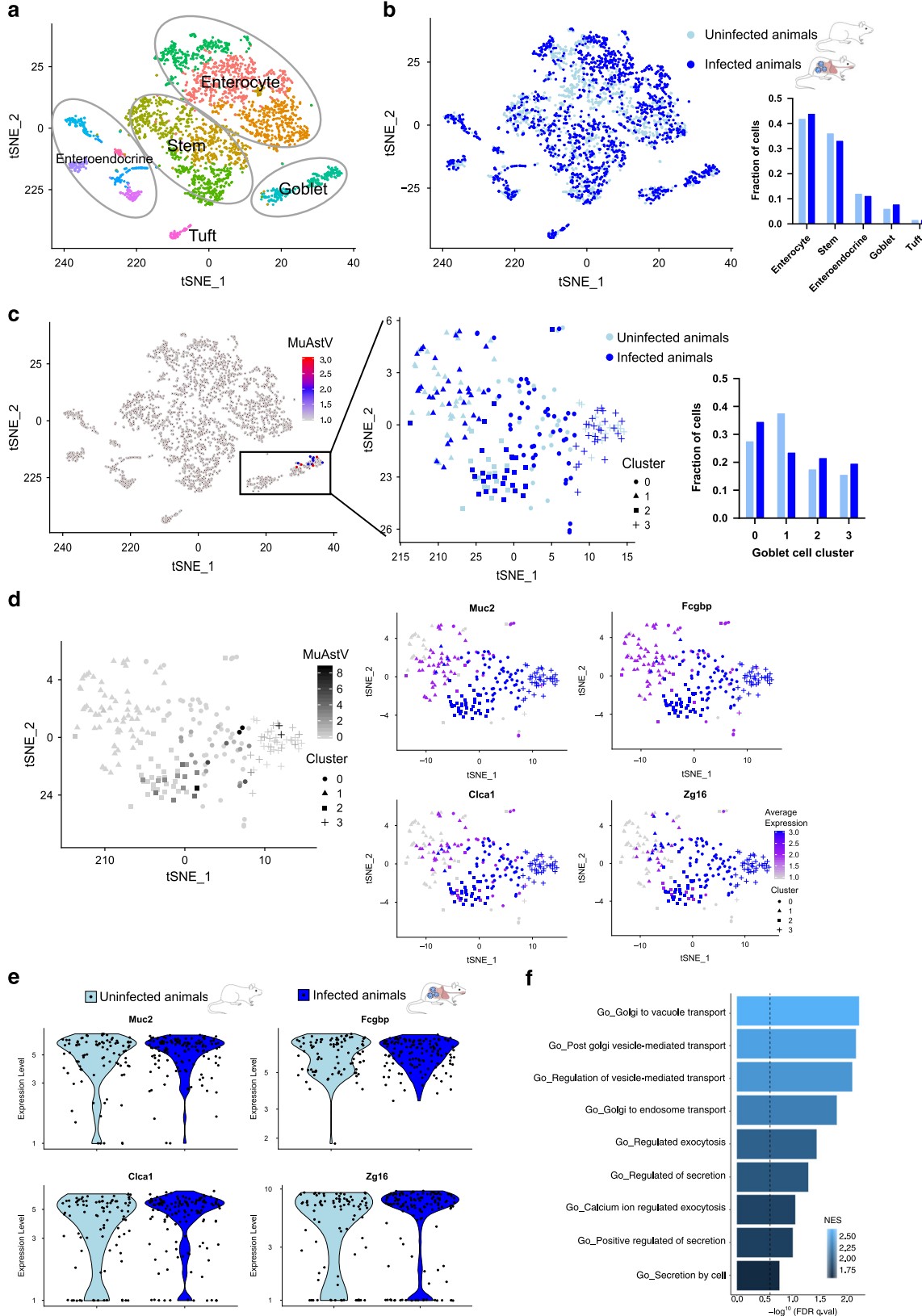

production via stimulation of the aryl hydrocarbon receptor on goblet cells can protect animals from EPEC infection[26] and also previous studies in Muc2-deficient animals whereby animals have increased morbidity and mortality after infection with *C. roden-tium*, a murine attaching/effacing bacterial pathogen[27]. Overall, these data reveal the functional consequences of MuAstV-

induced changes to the mucus barrier and demonstrate their ability to alter host susceptibility to infection.

In summary, these data represent significant evidence of enteric virus-infected goblet cells in vivo. These findings signify three major advances in our understanding of astroviruses. First, we have defined a main cell target of mammalian astroviruses that

**Fig. 2 Single-cell transcriptomics revealed murine astrovirus preferentially infects actively secreting goblet cells. a** Aggregated data of all duodenal epithelial cells (n = 2973 individual cells) from both infected and uninfected (n = 4 mice/group) as represented by t-SNE clustering and colored according to unique cell cluster based on transcriptional profiling. **b** Overlapping epithelial cell clusters from infected animals (dark blue) and uninfected animals (light blue) indicated that infection does not drive massive changes in epithelial cell populations. **c** Infected cells with detectable murine astrovirus are highlighted within the black box that includes all goblet cells. Subset analysis on goblet cells alone showed overlapping clusters of cells from infected (dark blue) and uninfected (light blue) animals. t-SNE clustering indicated four subpopulations of goblet cells as indicated by shape of each dot. **d** Cell clusters susceptible to infection are noted by overlay of murine astrovirus expression in grayscale. Murine astrovirus-susceptible cell clusters overlap with cells expressing high levels of mucus-related genes (*Muc2*, *Fcgbp*, *Clca1*, *Zg16*) as noted by darker colors. **e** Violin plots of mucus-related gene expression compared across infected (dark blue) and uninfected animals (light blue). **f** Gene set enrichment analysis indicating secretion pathways were significantly enriched in infected cells compared with uninfected cells using a weighted Kolmogorov–Smirnov-like statistic. Significance was estimated using an empirical phenotype-based permutation test and adjusted for multiple comparisons by first normalizing the gene set enrichment scores and calculating the false discovery rate (FDR) for each. Darker colors denote higher normalized enrichment scores (NES) for each gene set on the y-axis and an FDR q value < 0.2 is noted with a dotted line on the x-axis.

is consistent with the recent observation that goblet cells are infected by the human astrovirus strain, VA1, when using an enteroid model[28]. While enterocytes and progenitor cells also appeared to be infected by VA1, it was not determined which of these cell types were producing progeny virions[28]. This is an important point to consider given that we detected low levels of viral transcripts in nongoblet cell types, possibly indicating abortive infections (Supplementary Fig. 2). Alternatively, the detection of viral transcripts in nongoblet cells, which notably did not include all three viral ORFs, could represent an experimental artifact from scRNA-seq, whereby partitioning of cells could have inadvertently captured free-floating virus particles and/or viral RNA with uninfected cells. Instead, our data support productive infection as evidenced by high levels of all three ORFs in goblets cells expressing high levels of mucus-related genes, indicating that actively secreting goblet cells are a major site for replication. Thus, primary epithelial cell cultures that induce robust mucus secretion will be useful for future investigations in vitro. Second, our study indicates that mucus secretion increased after infection, which could stem from an increase in secretion within actively secreting cells as well as virus-induced changes to the goblet cell physiology that drives a more secretory phenotype. Upregulation of the secretory pathway is considered a host defense to expulse enteric pathogens[29,30], and a recent report showing that enterovirus 71 infection within goblet cells in enteroids actually reduces *Muc2* expression[31], likely to counteract this host response. Here we show coclustering of virus particles with mucus granules by electron microscopy (Fig. 1b), begging the question of whether the virus hijacks the mucus secretory pathway for egress. Indeed, many viruses have been shown to use host secretory pathways for nonlytic egress[32,33] and future studies are needed to define whether astrovirus can utilize the same secretory machinery as mucus. It is intriguing to consider that this host response that helps prevent damage to the gut epithelial barrier[9] could also function to promote virus replication. Finally, these studies highlight the role enteric viruses play in regulating the mucus barrier via goblet cells in the small intestine, which could translate into significant alterations in homeostasis, immune tolerance, and protection from gut pathogens[9]. This regulation could even be far-reaching throughout the gastrointestinal tract as we showed here with changes to the fecal microbiome, which is a summation of the overall gut microbiome, and reduced susceptibility to EPEC, an attaching/effacing bacterial pathogen. Additional studies are needed to examine the precise factors that mediate protection from EPEC, as both the increase in mucus and altered microbial communities could be at play.

In closing, MuAstV is endemic across all animal facilities tested[34–36], especially in immunocompromised mice that are incapable of clearing the virus. In fact, it was recently shown that

MuAstV infection can protect immunodeficient mice from murine norovirus infection via the induction of interferon lambda[37]. Therefore, these studies indicate that MuAstVs can potentially alter other mouse models of disease, including those focused on the microbiome as we showed here, and therefore are of importance to the biomedical research community.

## Methods

**Ethics**. All animal experiments were approved by the St. Jude Children's Research Hospital (St. Jude) Institutional Animal Care and Use Committee (protocol 570). St. Jude is fully accredited by the Association for the Assessment and Accreditation of Laboratory Animal Care International (AAALAC-I) and has an approved Animal Welfare Assurance Statement on file with the Office of Laboratory Animal Welfare (A3077-01). These guidelines were established by the Institute of Laboratory Animal Resources and were approved by the Governing Board of the U.S. National Research Council.

**Preparation and quantitation of virus inoculum**. Inoculum was prepared using homogenized intestines or feces that were centrifuged at 14,000 rpm for 2 min and then 0.22 μm-filtered[5]. To quantitate virus levels, RNA was extracted using the MagMAX-96 AI/ND Viral RNA Isolation Kit (Thermo Fisher Scientific) for feces and filtrates or the MagMAX Pathogen RNA/DNA Kit (Thermo Fisher Scientific) for tissues using the KingFisher Flex Purification System (Thermo Fisher Scientific). For experiments using DSS, lithium chloride precipitation solution (Thermo Fisher Scientific) was used according to the manufacturer's guidelines in order to remove any remaining DSS from RNA samples and prevent inhibition of downstream PCR assays. Copies of the MuAstV genome were quantified using a G-block standard (Integrated DNA Technologies) in a one-step qRT-PCR using TaqMan Fast Advanced Master Mix Virus (Applied Biosystems) with primers (F: TACATCGAGCGGGTGGTCGC, R: GTGTCACTAACGCGCACCTTTTCA) and probe ((6-FAM)-TTTGGCATGTGGGTTAA-(MBGNFQ) under the following conditions: 50 °C for 5 min, 95 °C for 20 s followed by 40 cycles of 95 °C for 3 s and 60 °C for 30 s on a BioRad CFX96 Real Time System[5,38,39].

**Animal experiments**. Adult (7-week-old) male and female wild-type C57BL/6 mice were purchased from The Jackson Laboratory and used 1 week after arrival for all experiments in adult animals. Original C57BL/6 breeders for neonate experiments were also purchased from The Jackson Laboratory. For all experiments, before inoculation on day 0, co-housed mice were confirmed to be negative for MuAstV by qRT-PCR screening of fresh feces. In all neonate experiments, breeding pairs were screened for MuAstV prior to experimentation as well as on day 0 of the infection. In cases where the number of neonates was unequal between MuAstV and mock infection groups, pups were redistributed on postnatal day 4 in order to have comparable numbers. For all experiments in adult animals, mice aged 8 weeks were orally inoculated with 100 mg/mL filtrate in 100 μL. For experiments in neonatal animals, 7-day-old mice were given 1 μL of filtrate. Mock infections used PBS alone. Samples of fresh feces and tissues were obtained at the time points indicated and stored at −80 °C until processed. For DSS experiments, drinking water containing 2% DSS (MP Biomedicals) was given to mice 1 day prior to infection with MuAstV. The DSS-containing water was changed every 2–3 days. After 7 days, animals were returned to normal drinking water. A dose of 0.2 μg of IL-4 (Sigma) complexed with 1 μg of functional grade anti-mouse IL-4 (clone 11B11; eBioscience) was given to each animal. Both complexed IL-4 and 0.2 μg of IL-13 (Sigma) were diluted in PBS for delivery via intraperitoneal injection given every other day for 10 days and a total of five injections, beginning 3 days prior to infection. Complexed IL-4 was also given to animals every other day starting after virus levels plateau (21 dpi) for 3 days and a total of two injections. For antibiotic experiments, mice were orally gavaged every other day over 5 days with a cocktail of vancomycin, ampicillin, neomycin, and metronidazole (10 mg of each/mouse)

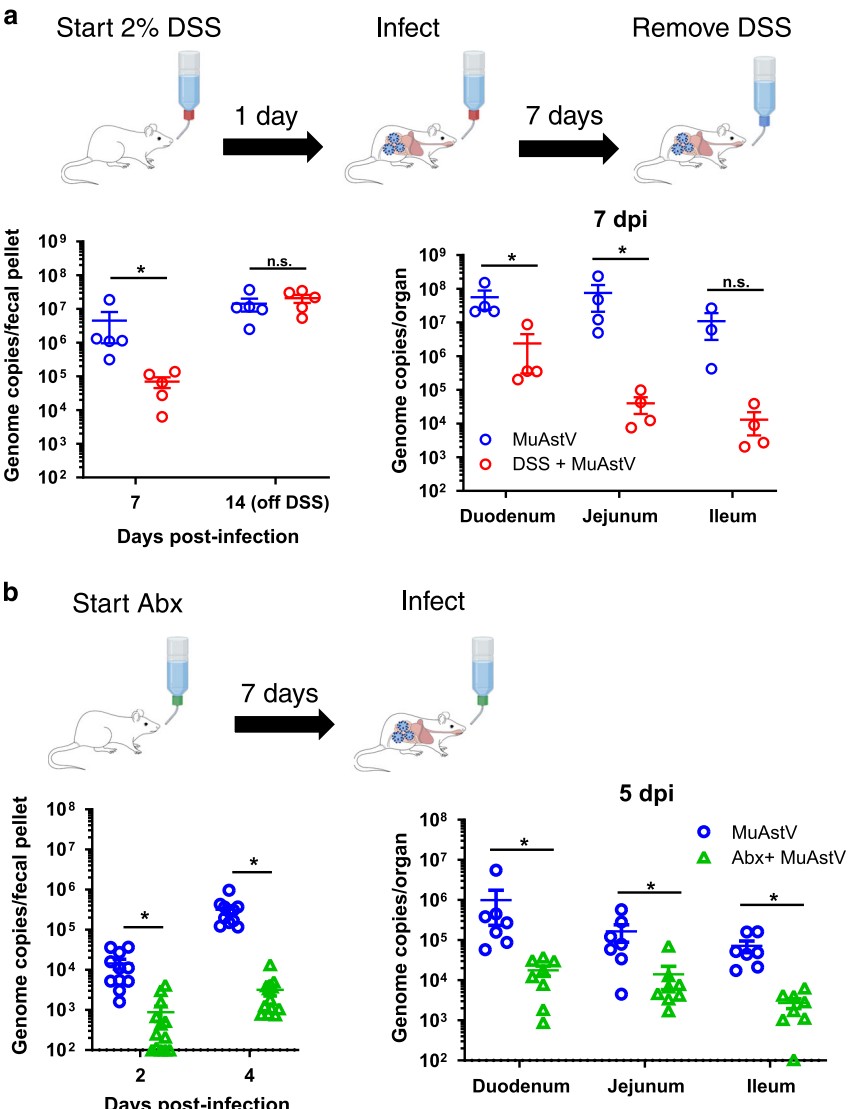

**Fig. 3 Disruption of mucus secretion resulted in reduced virus infection and shedding. a** 8-week-old mice were treated with 2% DSS for 1 day prior to infection and removed at 7 dpi. Treated mice (DSS + MuAstV; $n = 5$) shed significantly less virus in comparison with untreated mice (MuAstV; $n = 5$) ($p = 0.0079$), which was consistent with significantly reduced infection ($n = 4$/group) in the duodenum ($p = 0.0286$), jejunum ($p = 0.0286$), and a trending reduction in the ileum ($p = 0.0571$) at 7 dpi. Note one ileum sample from the untreated group is missing due to poor RNA yield. Levels of virus shed in the feces was not significantly different at 14 dpi ($p = 0.5476$), corresponding to 7 days after returning the DSS-treated group to water. Data shown are from a single experiment. **b** 8-week-old mice were treated with broad-spectrum antibiotics for 7 days prior to infection. Bacterial clearance from the gut was confirmed by fecal homogenates cultured on blood-agar plates incubated for 2 days under anaerobic conditions. Treated mice (Abx + MuAstV; $n = 13$) shed significantly less virus in comparison with untreated mice (MuAstV; $n = 11$) at 2 dpi ($p < 0.0001$) and 4 dpi ($p < 0.0001$), which was consistent with significantly reduced infection in the duodenum ($p = 0.0003$), jejunum ($p = 0.0140$), and ileum ($p = 0.0003$) from treated ($n = 8$) and untreated mice ($n = 7$). Data shown are from two independent experiments. For both panels, mean and standard error of the mean (SEM) are noted for each group. Y-axes are drawn at the lower limit of detection for the qRT-PCR assay. Significant differences between groups tested by a two-sided Mann–Whitney $U$ are noted (*). Abx antibiotics. Source data are provided as a Source Data file.

with cage changes after the first and last gavage. For the duration of the experiment, mice were given drinking water containing vancomycin (500 mg/L), ampicillin (1 g/L), neomycin (1 g/L), and metronidazole (1 g/L). Feces were homogenized using 1 mm glass beads and tenfold dilutions were made in PBS. Dilutions from 1:10[1] to 1:10[6] were plated on blood-agar plates and cultured in anaerobic chambers for 2 days. The absence of bacterial colonies was used to confirm clearance of culturable gut bacteria. Feces from untreated animals was used as a positive control. For experiments using EPEC (Xen-14, PerkinElmer), 7-day-old mice were mock or MuAstV-infected and 3 days later orally inoculated with $2.15-5.50 \times 10^5$ colony-forming units (CFU)/µL from overnight cultures of EPEC. Small intestines were collected in 0.5 mL of PBS and homogenized with either a hand blender or 1 mm glass beads (NextAdvance) in a Bullet Blender Tissue Homogenizer (NextAdvance) for 8 min at level 8 before tenfold dilutions were plated on MacConkey agar (Becton Dickson and Company) and CFUs were quantified the following day.

**Histopathology.** Mouse organs were formalin-fixed or fixed using Carnoy's solution (ethanol 6: glacial acetic acid 3: chloroform 1, v/v/v) for 2 h at 4 °C before being placed in 100% ethanol in order to preserve the mucus layer. Preserved tissues were embedded in paraffin, sectioned, and stained with a customized ISH probe specific for the MuAstV genome (Advanced Cell Diagnostics), periodic acid-Schiff stain or anti-Muc2 rabbit polyclonal antibodies (Abcam) at a 1:100 dilution to measure the mucus layer, and anti-DCLK1 rabbit polyclonal antibodies (Abcam) at a 1:100 dilution to identify tuft cells by the St. Jude Veterinary Pathology Core. The mucus layer at the top and in between villi was measured in at least three images per animal and quantified using the measurement tool in Image J software.

**Electron microscopy.** Mouse duodenums were fixed in a mixed aldehyde fixative in cacodylate buffer. Tissues were contrasted with osmium tetroxide and uranyl acetate, dehydrated in an ascending series of alcohols, transitioned in propylene

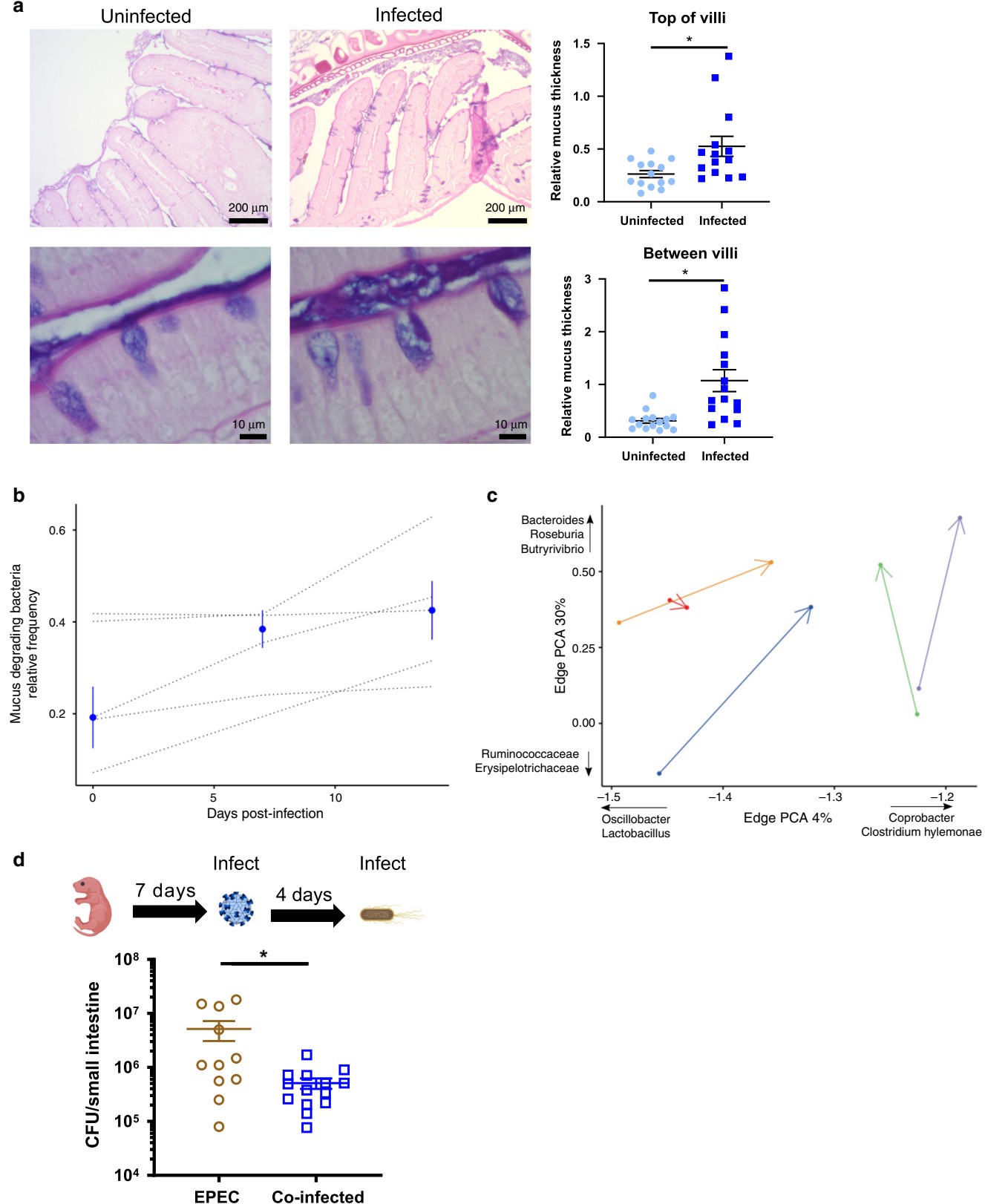

oxide and embedded in EmBed812 epoxy resin (Electron Microscopy Sciences). Embedded tissues were sectioned to glass slides and stained with toluidine blue to aid in final tissue selection. Selected areas were sectioned at ~80 nm and sections picked up on copper parallel bar grids. Samples were examined on a Thermo Fisher Scientific F20 transmission electron microscope operating at 80 kV and imaged with an AMT side-mount camera system.

**Single-cell RNA sequencing**. Duodenums were collected from infected and uninfected C57BL6 mice at 6 days post MuAstV infection ($n = 4$/group). Intestines were harvested on ice, washed twice with PBS, and then placed in stripping buffer (Hanks' Balanced Salt Solution (HBSS)), 10% fetal bovine sera, 15 mM HEPES, 5 mM EDTA, 5 mM DTT) for 30 min at 37 °C. A villus-targeted approach was taken to enrich for infected cells that were visualized by ISH only within the villi and not

**Fig. 4 Astrovirus-induced mucus secretion increased mucus-associated bacteria and enhanced resistance to enteropathogenic E. coli. a** Mucus thickness was visualized in small intestines collected at 7 dpi from infected and uninfected 8-week-old C57BL/6 mice (n = 5 animals/group) by periodic acid-Schiff staining and relative mucus thickness measured from three comparable regions for each animal in both groups. The mucus layer was thicker in infected animals at the top (p = 0.0082) and between (p = 0.0002) villi in comparison with uninfected animals The representative images of infected and uninfected animals are shown. Significant differences between groups were tested by a two-sided Mann–Whitney U are noted (*). Mean and SEM are noted for each group. **b** Relative frequency of well-characterized mucus-degrading bacteria (Supplementary Table 2) measured by 16S metagenomic sequencing of the fecal microbiome significantly increased after infection (linear mixed model; p < 0.001). Dotted lines represent the trends for each animal (n = 5) with median and standard deviation shown in blue. **c** Distinct microbiome shifts were observed between 0 (arrow origin) and 14 days post infection (arrowhead) as determined by phylogenetic edge principal component. Data shown for (**a–c**) are from a single experiment. **d** Groups of 7-day-old pups were mock infected (n = 11) or murine astrovirus (n = 11) infected before they were orally inoculated with EPEC at 3 dpi. Bacterial load was significantly lower in the small intestines of coinfected animals (p = 0.0172) harvested 4 days post EPEC infection, corresponding to 7 days post murine astrovirus infection. Significant differences between groups tested by a two-sided Mann–Whitney U are noted (*). Mean and SEM are noted for each group. Data shown are from two independent experiments. CFU colony-forming units. Source data are provided as a Source Data file.

the crypts. A downside of this approach was that it reduced the number of cells analyzed downstream because postmitotic differentiated cells from the villi undergo anoikis[40]. Digested intestines were filtered through 100 μm cell strainers and washed with cold PBS. Red blood cell lysis (Tonbo) was performed briefly, and cells were resuspended at $1 \times 10^6$ cells/mL. Cells were incubated with 1:100 dilutions of Ghost Dye Violet 510 (Tonbo), PE anti-mouse EPCAM (clone G8.8, Biolegend), and BV785 anti-mouse CD45.2 (clone 104, Biolegend) for 20 min at room temperature. Live CD45− EPCAM+ cells were sorted into 1.7 mL microfuge tubes containing HBSS and then centrifuged at $400 \times g$ for 5 min. To label cells from each mouse within the infection groups and enable multiplexing, cells were incubated with custom-designed hashtag oligos (HTO) conjugated to anti-PE antibodies (Thunder-Link PLUS Oligo Conjugation System, Expedeon). After washing and pooling across animals within each group, ~25,000 cells were loaded onto the Chromium controller (10x Genomics) to partition single cells into gel beads. Single-cell transcriptomic libraries were generated using the 5′ Gene Expression Kit (V2, 10x Genomics) according to the manufacturer's instructions with the addition of primers to amplify HTOs during cDNA amplification. Libraries were sequenced on the Illumina NovaSeq, generating ~500 M reads per sample.

**Single-cell gene expression analyses**. 10x gene expression data were first processed using Cell Ranger (v3.0.2, 10x Genomics) using a customized version of the mm10 reference that also included the complete MuAstV genome (GenBank accession JX544744.1). Samples were aggregated and normalized by the median number of mapped reads per identified cell. Normalized feature-barcode matrices including both gene expression and HTO counts were then imported into Seurat (v3.0.0.900) for downstream analysis and data visualization. Data were first filtered by excluding any gene that was not present in at least 0.1% of total called cells and then by excluding cells that exhibited extremes in the distributions of the number of genes expressed (<500 or >5000), the number of mRNA molecules (mouse: >25,000), or the percent of expression owed to mitochondrial genes (>10%). Gene expression counts were log-normalized using a scaling factor of 1e4, variable features targeted for downstream analysis were identified using the "vst" selection method with default parameters (excluding astrovirus ORFs), and cell cycle scores were generated using for each cell using markers identified elsewhere. Gene expression was then scaled to regress out the effects of total transcript expression, percent of mitochondrial expression, and inferred cell cycle scores. For first-pass analyses of entire datasets, we selected the first 21 PCs visually using an elbow plot of the PC standard deviations. These PCs were then used for downstream t-SNE dimensionality reduction and cell clustering with Seurat's shared nearest neighbor modularity algorithm in order to broadly characterize the cell types of each sample. Individual cell subsets were annotated using known markers from the literature[40].

For more detailed analyses of goblet cell populations, we identified the top variable features within this subset, and rescaled that data as described above. After reconducting PC analysis, PCs were scored for significance (FDR-adjusted p value < 0.05) using random permutations as implemented in Seurat, resulting in the use of the first six PCs for downstream analyses. Differential gene expression was assessed among clusters for all genes expressed in at least one percent of cells within a cluster using a generalized linear hurdle model that incorporates both expression frequency and abundance. For pairwise cluster comparisons of interest, genes were then ranked as a function of the product of their average log fold change, the absolute value of the difference in percent expression, and the inverse of the scaled, FDR-adjusted p value. These gene rank lists were then analyzed using preranked gene set enrichment analysis.

**Microbiome**. DNA was extracted from fecal pellets collected at 0, 7, and 14 dpi using methods to improve the bacterial species captured in a sample[41,42] before 16S rRNA gene amplicon sequencing was performed[43] at the St. Jude Hartwell Center. Briefly, Illumina MiSeq paired-end reads were run through DADA2 pipeline[44] (version 1.10.1) to correct sequencing errors and determine amplicon sequence

variants (ASVs), which represent original 16S rRNA gene amplicons in the samples. The ASVs were then used to recruit full-length 16S rRNA gene sequences from Ribosomal Database Project release 16.0[45] to construct a phylogenetic reference dataset from which a reference tree was built. The amplicon sequences were then placed onto the reference tree using the pplacer tool[46]. Microbial diversity was measured by three separate indices: Simpson (nonphylogenetic measure to classified operational taxonomic units), phylogenetic quadratic entropy (phylogenetic generalization of the Simpson) and balanced weighted phylogenetic diversity[47] (incorporates proportional abundance to phylogeny). The relative frequency of mucus-degrading bacteria in the fecal microbiome of mice was determined from a database of well-described species (see Supplementary Table 1)[22,23] and kinetics analyzed with linear mixed models. Shifts in the microbiome were determined by phylogenetic edge PCs analysis[48].

**Reporting summary**. Further information on research design is available in the Nature Research Reporting Summary linked to this article.

## Data availability
scRNA-seq and 16S metagenomic sequencing data that support the findings of this study have been deposited in NCBI BioProject with the primary accession code PRJNA573959. Reference MuAstV genome is available through GenBank accession JX544744.1. Data underlying Figs. 3a, b, 4a, d, and Supplementary Fig. 5a, c are provided as Source Data files. All other data are available from the corresponding author upon reasonable requests.

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

## Acknowledgements

The authors thank the members of the St. Jude Animal Resource Center, Flow Cytometry Core, Cell and Tissue Imaging Center, Hartwell Center, Veterinary Pathology Core, and Aaron Poole for technical assistance. The authors also thank Rebekah Honce for graphical design. Funding for this research included National Institutes of Health Allergy and Infectious Diseases grants R01 AI121832 (P.G.T.), R01 AI136514 (P.G.T.), R21 AI135254-01 (S.S.-C.), R03 AI126101-01 (S.S.-C.), Primary Caregiver Technical Assistance Supplement (V.C.), T32 AI106700-03 (V.C.), as well as funding from ALSAC (S.S.-C.). The Cell & Tissue Imaging Center was supported by in part by funding from the National Cancer Institute P30 CA021765.

## Author contributions

Conceptualization: V.C., D.F.B., P.G.T., and S.S.-C. Data curation: V.C. and J.C.C. Formal analysis: V.C., J.C.C., H.R., R.A., and E.M. Funding acquisition: V.C., C.G.R., P.G.T., and S.S.-C. Investigation: V.C., D.F.B., B.S., B.L., H.R., A.D., B.S., and P.V. Methodology: V.C., D.F.B., J.C.C., H.R., J.R., E.M., P.G.T., and S.S.-C. Project administration: V.C., C.G.R., P.V., J.R., E.M., P.G.T., and S.S.-C. Resources: V.C., D.F.B., J.C.C., B.S., B.L., H.R., R.A., C.G.R., P.V., J.R., E.M., P.G.T., and S.S.-C. Software: V.C., D.F.B., J.C.C., R.A., and E.M. Supervision: V.C., C.G.R., P.V., J.R., E.M., P.G.T., and S.S.-C. Validation: all authors. Visualization: V.C., D.F.B., J.C.C., H.R., B.W., R.A., C.G.R., P.V., and E.M. Writing-original draft: V.C. Writing review and editing: all authors.

## Competing interests

The authors declare no competing interests.
