## [Peer Review File · Nature Communications]

Reviewers' comments:

Reviewer #1 (Remarks to the Author):

In the manuscript by Cortez et al., the authors demonstrate for the first time that goblet cells are a target for murine astrovirus and that viral infection influences mucus production and intestinal microbial niches. The authors use multiple approaches including light microscopy, EM, and single-cell RNA sequencing to demonstrate astrovirus tropism for goblet cells. Further, the authors underscore an indirect and novel transkingdom interaction between astrovirus and mucus-associated bacteria, bolstering colonization resistance by a pathogenic *E. coli* strain. This is an interesting and impactful study because the cell tropism and consequences of murine astrovirus have long been unknown. This work has critical implications for studies of human astrovirus and more broadly in our understanding of enteric viral pathogenesis. The findings described herein are both novel and impactful and will substantially advance the field.

Major points:

1. The authors demonstrate two methods to deplete mucus which correlate with reduced viral loads; however, the reverse is also important. Therefore, an orthogonal approach showing that increased mucus increases viral infection (such as by IL-4, IL-13, or IL-25 treatment of mice) will strengthen their model.
2. The authors claim astrovirus selectively infects actively secreting goblet cells. There are alternative hypotheses such as virus re-programming of goblet cells or virus differentially killing different goblet cell types that should be considered. This is likely not yet possible to reconcile experimentally given the limited tools to study both astrovirus and goblet cells, but the text should better reflect the uncertainty between correlation and causation underlying the observed preference for actively secreting goblet cells.

Minor points:

1. A negative (uninfected) control would be of value in Figure 1. Ideally, immunofluorescence microscopy demonstrating colocalization of an astrovirus non-structural protein with a goblet cell marker such as Muc2 would be included since current evidence is limited to viral RNA and EM.
2. Figure 2B should be clarified, as current coloration suggests many cell types are infected. I assume the authors suggest the blue dots are ALL cells (infected and uninfected) from infected mice rather than simply infected cells from infected mice.
3. The authors figures do not adhere to the Nature checklist. For instance, scale bars are needed for all images. Fig S1 should have antigens/colors identified. What are the red dots in Fig 2C?

4. How many infected cells are in each of the 5 cell populations identified in Figure 2A? Are there zero infected cells in all non-goblet populations?
5. How does the cellular composition compare in infected and uninfected mice in Fig 2B? Quantitation of cell numbers would be helpful in figure or text to demonstrate whether astrovirus infection does not induce changes to epithelial composition. The authors state there are no significant changes but including numbers would be helpful.
6. In Figure 2E it would be helpful to also include a merge of average expression between the infected and uninfected cells in mice administered astrovirus for more accurate comparison with the uninfected mouse.
7. Methods should be clarified: Were littermates used? Were mice singly-housed or co-housed? How was virus stock made and tittered for animal inoculations?
8. Several typos: (“uninfected” in Figure 2E, “tuft cell maker in Figure S1, etc.)

Craig Wilen

Reviewer #2 (Remarks to the Author):

In this study, the authors Cortez et al use their recently characterized mouse model of murine astrovirus infection to better understand the pathogenesis of this relatively poorly understood enteric virus. Using combined techniques in in-situ hybridization with a viral-targeting probe, transmission electron microscopy, and single-cell RNA seq, the authors convincingly pinpoint the cellular tropism of the virus to a subset of small intestinal goblet cells (in duodenum and jejunum) that have a secretory phenotype. The secretions from these goblet cells also seem to modulate the mucus layer throughout the small intestine, which impacts the microbiota by increasing numbers of mucus-associated bacteria, and the mucus prevents EPEC infection in neonatal mice. Therefore, it is concluded that this new model system reveals secretory goblet cells of the small intestine to be targets for infection and replication for Astrovirus that impacts the gut ecosystem and protection from enteric bacterial infection. This is overall a well-written and executed study with sound experimental approaches that offer new insights into Astrovirus infection that will be welcomed in the field. That said, there are several key points/caveats that need to be addressed before it should be accepted for publication.

Major Points

1. I have two major concerns with the conclusion that the virus “hijacks” specifically secretory goblet cells. First, the data are not conclusive that the virus targets goblet cells that are functionally more secretory vs. goblet cells that are not. An alternative explanation is that the subset of goblet cells that the virus targets have other properties that guide the viral tropism, and that the secretory response is secondary to the infection once the goblet cells sense the virus. For example, it is known that a subset of goblet cells harbor innate receptors (eg. NLRP6) that can sense bacterial or metabolite ligands and promote a rapid secretory response (PMID: 27339979). Besides the secretion related genes, were any viral or DNA-sensing pattern recognition receptors expressed in the infected goblet cells as described by RNASeq? What other genes set the infected vs. uninfected duodenal and jejunal goblet cells apart?

2. Second, while the authors perform functional experiments (DSS treatment and ABX treatment) to prevent goblet cell secretion to prove their hypothesis, these are not direct enough to address that question, since DSS is cytotoxic and therefore impacts many cell functions, and ABX will also cause many changes to the epithelium (PMID:15260992). At the very least, to complement the PCR assays in Fig S3, the authors should show H&E, viral infection by ISH, and mucin staining (PAS or Muc2) on the Vehicle vs. DSS (Fig. S3A) and ABX-treated (Fig. S3B) small intestine to characterize the effects of the interventions on gut morphology, and confirm both these functional interventions reduce mucus secretion and viral tropism to goblet cells. Further, as a more direct study, the authors may want to try the model in *Agr2*^{-/-} mice (available from Jackson lab #025630) where goblet cells cannot produce Muc2, or think of an alternative assay by blocking secretion in mucosal explants as frequently done by Hansson and colleagues (again, PMID: 27339979). Due to the direct relationship of these studies to the hypothesis (the virus targets secreting goblet cells), this data should also be a main figure, not a supplementary.

3. The authors show the increased mucus production impacts the mucus- associated bacteria, but the microbiota analysis is evidently done on fecal samples. How do the authors speculate the small bowel mucus impacts the colon microbiome? The authors should also mention in discussion that the impact of the virus on the neonatal EPEC infection may also be secondary to the changes in the microbiota.

4. It is evident that previous work including the first and senior authors has characterized the MuAstV model (PMID: 30971471), but this should be better clarified/summarized in the main text. As it stands, the current text vaguely refers to the establishment of a murine astrovirus model in a previous study (Line 29 – 31), but implies the use of MuAstV is unique to this study (Line 37), especially in abstract mentioning the authors have in this study “developed a murine model to study astrovirus pathogenesis”.

5. Although the authors do a convincing job showing goblet cell tropism, it is difficult to envision how the virus is able to infect the goblet cell. For example, the ISH in Fig 1A shows the signal mostly in the basolateral region, which may explain how the virus replicates, but the SEM (Fig. 1B) shows the virus in the goblet cell theca amidst the granules. How do the authors explain how the virus gets into the basolateral region, make it to the theca, and reinfect new goblet cells?

Minor Points

1. Aside from one panel in Fig S5, there are no clear negative controls for the virus probe, e.g. a non-related viral probe (rotavirus), or non-gut related tissue of an infected mouse (e.g lung). These would help more clearly indicate specificity.

2. The authors claim that the virus can be found on colon mucus, but the data used to demonstrate this more looks like the glycocalyx. What part of the colon is analyzed here (looks like proximal colon). Is the virus also in distal colon mucus? Does it interact with bacteria?

3. EPEC is an attaching/effacing pathogen with a murine counterpart *Citrobacter rodentium*. With the in vivo findings that the increased mucus reduces susceptibility to EPEC infection, the authors should put their findings in context of previous studies looking at the in vivo role of the Muc2 mucin in protection from *C.rodentium* infection (PMID:20485566)

Reviewer #3 (Remarks to the Author):

In the manuscript "Astrovirus hijacks goblet cell secretory pathway for replication and alters gut mucus barrier" Cortez et al investigate the tropism of murine astrovirus infection. Cortez et al demonstrate a specific staining pattern of astrovirus infected cells consistent with goblet cells rather than bulk infection of all intestinal epithelial cells. Furthermore, Cortez et al perform single cell sequencing to further define a subset of goblet cells that are infected. Cortez et al interrogate whether mucus production is altered during murine astrovirus infection and the consequence that this has on the microbiome and bacterial infection. The general findings are of great interest;

however, the claims of viral hijacking of the secretory pathway are overstated. While this reviewer appreciates the difficulty in interrogating goblet cell function in vivo, the assays performed (DSS and Antibiotic treatment) are too broad and pleiotropic to confidently describe a co-option or hijacking as claimed in the text. Additionally, the scRNA seq and 16S analysis needs to be further expanded to better illustrate the authors points.

Major Points

1. The authors claim in the title and throughout the text (e.g. line 177) that astrovirus co-opts secretory pathway for replication. This statement is too strong given the evidence presented. The authors should consider rewording these statements or provide substantially more evidence to substantiate this claim. These experiments would need to include alterations in mucus production through cytokines (e.g. IL-4) and mouse genetics while quantifying viral infection and goblet cell numbers (which are not done in the DSS or antibiotic experiments currently). Additionally, the authors would also need to provide more direct evidence through an in vitro culturing system such as enteroids that mucus enhanced viral replication.
2. The authors should show a co-stain with Muc2 or another Goblet cell marker in Figure 1. Additionally, this should be quantified to demonstrate which percentage of the stained cells are goblet cells.
3. The authors claim that there are 4 different goblet cell groups but it isn't clear what markers and signatures define these populations. The authors should include further analysis demonstrating which genes for goblet cells are associated with the different subpopulations similar to what has been done for enteroendocrine and tuft cell subpopulations in the paper by Haber et al Nature (PMID: 29144463). This analysis will be useful for future investigations into astrovirus infection of goblet cells.
4. The authors claim that the only 3 of 4 populations are susceptible to MuAstV infection (line 76-77). However, this reviewer questions whether or not the data is robust enough to make this claim based on scRNA sequencing alone. For example, are enough reads in the 1 population of cells sufficient to exclude infection that was missed by the depth limitations of scRNA seq?
5. The 16S data needs more analysis to demonstrate the specific changes in the microbiome. For example, the authors should present the relative abundance order/family/genus 16S rDNA sequence assignments in a bar chart (or similar graph) of the major bacterial communities. Additionally, it will be important to note which taxa are discriminate? Is it only the mucus associated bacteria? Does the diversity change with astrovirus infection? These analyses will help put the mucus associated bacteria data in better context.
6. The authors should note that changes in the microbiome by MuAstV (which may or may not be dependent upon mucus changes) may also contribute to the EPEC infection phenotype.

Minor Points

1. The text and figures should be expanded to add clarity from its current short format.

2. It appears from the scRNA sequencing that only goblet cells are infected by murine astrovirus, but didn't appear to be explicitly stated in the text. Did the authors detect MuAstV reads in other cell populations?
3. Figure 2 gets difficult to read at its current size and resolution. This is particular true of Figure 2D as it is difficult to see the symbols. Perhaps a violin plot may complement this point.
4. Text on Figure S4 is very difficult to read and interpret.
5. Does this work have any implications for vitro culturing for MuAstV?
6. The authors should discuss MuAstV tropism in light of the Human Astrovirus tropism in vitro findings by the Wobus lab (PMID: 31671153).

Reviewers' comments:

Reviewer #1 (Remarks to the Author):

In the manuscript by Cortez et al., the authors demonstrate for the first time that goblet cells are a target for murine astrovirus and that viral infection influences mucus production and intestinal microbial niches. The authors use multiple approaches including light microscopy, EM, and single-cell RNA sequencing to demonstrate astrovirus tropism for goblet cells. Further, the authors underscore an indirect and novel transkingdom interaction between astrovirus and mucus-associated bacteria, bolstering colonization resistance by a pathogenic *E. coli* strain. This is an interesting and impactful study because the cell tropism and consequences of murine astrovirus have long been unknown. This work has critical implications for studies of human astrovirus and more broadly in our understanding of enteric viral pathogenesis. The findings described herein are both novel and impactful and will substantially advance the field.

Response: We thank the Reviewer for their appreciation of these studies.

Major points:

1. The authors demonstrate two methods to deplete mucus which correlate with reduced viral loads; however, the reverse is also important. Therefore, an orthogonal approach showing that increased mucus increases viral infection (such as by IL-4, IL-13, or IL-25 treatment of mice) will strengthen their model.

*Response: We agree that an orthogonal approach would be appropriate and the use of cytokines to increase mucus production was also suggested by Reviewer #3. However, upon testing with both IL-4 and IL-13 treatments individually, we observed a reduction in virus shed (below, panel A) as well as virus infection in the tissue (below, panel B). This is consistent with what we observed by histology (below graphs), demonstrating that these treatments (administered I.P. on -3, -1, 1, 3, and 5 days post-infection) led to a reduction in goblet cells. A similar course of treatment was used to increase goblet secretion in the context of *C. rodentium* infection Sharba et al. Virulence 2019, but their focus was on colonic goblet cells.*

As a second approach, we treated animals with IL-4 after peak infection, when virus levels plateau; however, this also failed to increase virus shed in the feces on day 25 after 2 doses of IL-4 administered I.P. on days 21 and 23 (below, panel C). This method was employed by Wilen et al. Science 2018. However, they also noted a regional effect of IL-4 increasing tuft cells in the ileum, but not the colon of antibiotic treated animals. Thus, it is possible that the proximal regions of the duodenum and jejunum, where astrovirus preferentially replicates, is distinctive from these other regions of the GI tract. These may also point to pathogen-specific differences and while they are intriguing findings on their own, substantial investigations would be needed to follow up on these results, which we deem to be outside the scope of this study.

2. The authors claim astrovirus selectively infects actively secreting goblet cells. There are alternative hypotheses such as virus re-programming of goblet cells or virus differentially killing different goblet cell types that should be considered. This is likely not yet possible to reconcile experimentally given the limited tools to study both astrovirus and goblet cells, but the text should better reflect the uncertainty between correlation and causation underlying the observed preference for actively secreting goblet cells.

Response: We agree with the Reviewer and these alternate hypotheses have been considered. First, we do not believe our data support that the virus is differentially killing goblet cell types. This is based on the lack of significant differences in goblet cells and subpopulation between infected and uninfected animals. We have now made this point clearer in Figure 2c, third panel, which includes quantitation of the subpopulations and testing for differences using χ^2 analysis (line 83). Second, we do not think that targeting actively secreting cells and re-programming them to be more secretory are mutually exclusive. In fact, we believe our data support both scenarios and we show the former by scRNA-seq and functional analyses using DSS and ABX. To the latter point, we found that if we simply classified goblet cells as either secretory or non-secretory, we found a slight increase among infected animals (61%) in comparison to uninfected animals (52%) (line 98). While this difference was not statistically significant, we have now included these lines of thought in the Discussion (lines 208-211).

Minor points:

1. A negative (uninfected) control would be of value in Figure 1. Ideally, immunofluorescence microscopy demonstrating colocalization of an astrovirus non-structural protein with a goblet cell marker such as Muc2 would be included since current evidence is limited to viral RNA and EM.

Response: The uninfected controls from Figure S1a have now been moved to the main Figure 1a. Unfortunately, we do not yet have antibodies for the non-structural viral proteins. The procedures for Muc2 staining as well as alcian blue with the ISH probe was incompatible (now mentioned line 45), but we have now included Muc2 staining on a serial section for comparison as shown below at 3 resolutions and now included the 20X figures as the new Figure S1b. Finally, our data are not limited to ISH staining and EM, as we performed single-cell transcriptomics that bolster our findings.

2. Figure 2B should be clarified, as current coloration suggests many cell types are infected. I assume the authors suggest the blue dots are ALL cells (infected and uninfected) from infected mice rather than simply infected cells from infected mice.

Response: Yes, all light blue dots are from uninfected animals and all dark blue dots are from infected animals. We apologize for the confusion. We have now clarified this point in the text of the figure legend, as well as the figure labels.

3. The authors figures do not adhere to the Nature checklist. For instance, scale bars are needed for all images. Fig S1 should have antigens/colors identified. What are the red dots in Fig 2C?

Response: Thank you for this comment. Scale bars, antigens/colors in Figure S1 and expression scale in Figure 2c have now all been included.

4. How many infected cells are in each of the 5 cell populations identified in Figure 2A? Are there zero infected cells in all non-goblet populations?

Response: Based on scRNA-seq alone, this is a difficult question to address for 2 reasons. First, it is possible that virus particles or free-floating viral RNA could be sequestered with an uninfected cell during the partitioning step of the 10x Genomics pipeline:

10x Technology for Single Cell Partitioning

Second, viruses can enter a cell but then do not produce progeny virions, and we define this as an abortive infection. Unfortunately, this technology cannot distinguish between these 2 possibilities. However, based on the expression of viral transcripts across the 5 epithelial cell populations (new Figure S3), we do not believe there is productive replication in non-goblet cells because none of these cell populations express all 3 astrovirus ORFs. This is further supported by our ISH staining (Figure 1a and Figure S2) where we only observed robust staining in small intestinal goblet cells. These points of discussion are now included beginning on line 199.

5. How does the cellular composition compare in infected and uninfected mice in Fig 2B? Quantitation of cell numbers would be helpful in figure or text to demonstrate whether astrovirus infection does not induce changes to epithelial composition. The authors state there are no significant changes but including numbers would be helpful.

Response: We have now quantified the cells and placed these numbers next to Figure 2b. The numbers do not significantly differ between infected and uninfected animals (line 74).

6. In Figure 2E it would be helpful to also include a merge of average expression between the infected and uninfected cells in mice administered astrovirus for more accurate comparison with the uninfected mouse.

Response: We have replaced Figure 2e with violin plots (also requested by Reviewer #3) that better summarize the overall comparisons between infected and uninfected animals.

7. Methods should be clarified: Were littermates used? Were mice singly-housed or co-housed? How was virus stock made and tittered for animal inoculations?

Response: Mice were purchased directly from Jackson Laboratory and then used after a 1-week acclimation period. It is probable that many were littermates, but this was not a stipulation in our experiments. All mice were co-housed for experiments. We have now included these details in our Methods lines 254-257. Virus stocks were made as described in the Methods beginning on line 242.

8. Several typos: ("uninfected" in Figure 2E, "tuft cell maker in Figure S1, etc.)

Response: Corrected.

Craig Wilen

Reviewer #2 (Remarks to the Author):

In this study, the authors Cortez et al use their recently characterized mouse model of murine astrovirus infection to better understand the pathogenesis of this relatively poorly understood enteric virus. Using combined techniques in in-situ hybridization with a viral-targeting probe, transmission electron microscopy, and single-cell RNA seq, the authors convincingly pinpoint the cellular tropism of the virus to a subset of small intestinal goblet cells (in duodenum and jejunum) that have a secretory phenotype. The secretions from these goblet cells also seem to modulate the mucus layer throughout the small intestine, which impacts the microbiota by increasing numbers of mucus-associated bacteria, and the mucus prevents EPEC infection in neonatal mice. Therefore, it is concluded that this new model system reveals secretory goblet cells of the small intestine to be targets for infection and replication for Astrovirus that impacts the gut ecosystem and protection from enteric bacterial infection. This is overall a well-written and executed study with sound experimental approaches that offer new insights into Astrovirus infection that will be welcomed in the field. That said, there are several key points/caveats that need to be addressed before it should be accepted for publication.

Response: We thank the Reviewer for their appreciation of this important research topic.

Major Points

1. I have two major concerns with the conclusion that the virus “hijacks” specifically secretory goblet cells. First, the data are not conclusive that the virus targets goblet cells that are functionally more secretory vs. goblet cells that are not. An alternative explanation is that the subset of goblet cells that the virus targets have other properties that guide the viral tropism, and that the secretory response is secondary to the infection once the goblet cells sense the virus. For example, it is known that a subset of goblet cells harbor innate receptors (eg. NLRP6) that can sense bacterial or metabolite ligands and promote a rapid secretory response (PMID: 27339979).

Response: We agree that this is an interesting point to consider. As mentioned in response to Reviewer #1's Major Point #1, if the secretory response was solely secondary to infection, we would expect the proportion of secretory goblet cells between infected and uninfected animals to be different. The virus infects 43% of total goblet cells, but there is not a corresponding increase in secretory goblet cells in infected animals. Instead, based on our quantitation, each of the 4 subpopulations of goblet cells as well as total goblet cells remains comparable between infected and uninfected animals (Figure 2b and c, respectively). However, we acknowledge that the differentiation state and/or expression of a receptor may be another key distinguishing factor and this is under active investigation. We also acknowledge the alternative hypotheses proposed by Reviewer 1.

Besides the secretion related genes, were any viral or DNA-sensing pattern recognition receptors expressed in the infected goblet cells as described by RNASeq?

Response: We did not identify any differentially expressed pattern recognition receptors. However, infected cells expressed interferon stimulated genes (e.g. Isg15, Irf7, Ili27l2b) (Table S1) and were enriched for antiviral programming by gene set enrichment analysis in comparison to goblet cells from uninfected animals (In preparation). While it is unclear how these may be directly linked to active secretion, it is an intriguing possibility that can be the focus of future studies.

What other genes set the infected vs. uninfected duodenal and jejunal goblet cells apart?

Response: We have now included a list of the top differentially expressed genes between infected and uninfected cells within infected animals in the new Table S1.

2. Second, while the authors perform functional experiments (DSS treatment and ABX treatment) to prevent goblet cell secretion to prove their hypothesis, these are not direct enough to address that question, since DSS is cytotoxic and therefore impacts many cell functions, and ABX will also cause many changes to the epithelium (PMID:15260992). At the very least, to complement the PCR assays in Fig S3, the authors should show H&E, viral infection by ISH, and mucin staining (PAS or Muc2) on the Vehicle vs. DSS (Fig. S3A) and ABX-treated (Fig. S3B) small intestine to characterize the effects of the interventions on gut morphology, and confirm both these functional interventions reduce mucus secretion and viral tropism to goblet cells. Further, as a more direct study, the authors may want to try the model in Agr2^{-/-} mice (available from Jackson lab #025630) where goblet cells cannot produce Muc2, or think of an alternative assay by blocking secretion in mucosal explants as frequently done by Hansson and colleagues (again, PMID: 27339979). Due to the direct relationship of these studies to the hypothesis (the virus targets secreting goblet cells), this data should also be a main figure, not a supplementary.

Response: We have now included the requested H&E, ISH, and Muc2 staining for DSS and ABX experiments (Figure S4) and moved the ABX and DSS data to the main text (Figure 3). While we previously considered the use of Agr2^{-/-} mice based on Hansson et al., it was found in a subsequent publication that the mucus barrier is not disrupted in these mice and this is thought to be due to facility-to-facility differences (e.g. husbandry, microbiome) (Bergstrom PLOS One 2014). Thus, to embark on the rederivation of this mouse strain from

Jackson Laboratory would have potentially given us only a 50% chance of yielding biological insights and would have come at significant cost, both monetarily and in the delay of disseminating these findings. Still, we think the approach would be ideal if it was less a. Similarly, our laboratory does not have expertise in explant models, and thus, we are hesitant to embark on such an investment at this time. Instead, we are actively working on an in vitro system, as mentioned in more detail below in response to Reviewer #3's request for an enteroid model, as it will be key to furthering these studies.

3. The authors show the increased mucus production impacts the mucus-associated bacteria, but the microbiota analysis is evidently done on fecal samples. How do the authors speculate the small bowel mucus impacts the colon microbiome? The authors should also mention in discussion that the impact of the virus on the neonatal EPEC infection may also be secondary to the changes in the microbiota.

Response: The Reviewer is correct, the analysis was performed on fecal samples, which provides a summation of the microbiome in the GI tract. We believe this makes our data all the more striking and could potentially implicate downstream effects in distal sites of the GI tract. To what end those effects may translate to functional mucosal changes in the colon is a very intriguing question that will warrant subsequent study. We have included this possibility as well as remarked on the possibility that the reduced neonatal EPEC colonization in astrovirus-infected animals may be in part due to these changes to the microbiome in the Discussion (lines 222-227).

4. It is evident that previous work including the first and senior authors has characterized the MuAstV model (PMID: 30971471), but this should be better clarified/summarized in the main text. As it stands, the current text vaguely refers to the establishment of a murine astrovirus model in a previous study (Line 29 – 31), but implies the use of MuAstV is unique to this study (Line 37), especially in abstract mentioning the authors have in this study “developed a murine model to study astrovirus pathogenesis”.

Response: We apologize for this confusion as we did not mean to imply that this model was unique to this study, rather that it was just characterized only recently. We have altered the text to reflect this distinction in the abstract (line 15) and introduction (line 36).

5. Although the authors do a convincing job showing goblet cell tropism, it is difficult to envision how the virus is able to infect the goblet cell. For example, the ISH in Fig 1A shows the signal mostly in the basolateral region, which may explain how the virus replicates, but the SEM (Fig. 1B) shows the virus in the goblet cell theca amidst the granules. How do the authors explain how the virus gets into the basolateral region, make it to the theca, and reinfect new goblet cells?

Response: Astroviruses infect via the apical side of polarized epithelial cells. We agree with the Reviewer that the virus is replicating in the perinuclear region on the basolateral side of the goblet cell. The virus then traffics to the theca, as shown by EM and ISH (Figure 1a and b), for egress, which we believe supports our speculation that the virus is hijacking the secretory pathway of goblet cells. The virus released in the mucus is then capable of infecting new goblet cells from the apical side.

Minor Points

1. Aside from one panel of Fig S5, there are no clear negative controls for the virus probe, e.g. a non-related viral probe (rotavirus), or non-gut related tissue of an infected mouse (e.g lung). These would help more clearly indicate specificity.

Response: We have moved the uninfected stained with the ISH probe as negative controls to the main Figure 1 and now show a lack of staining in the lung of an infected mouse (Figure S1a), together indicating the clear specificity of this probe.

2. The authors claim that the virus can be found on colon mucus, but the data used to demonstrate this more looks like the glycocalyx. What part of the colon is analyzed here (looks like proximal colon). Is the virus also in distal colon mucus? Does it interact with bacteria?

Response: The virus can only be found in the proximal, but not the distal colon (now Figure S1f), which we have now included on line 53. We agree that the virus may be binding to the glycocalyx but would need higher resolution to determine that. We do not yet know of direct interactions with bacteria, but it is likely given the substantial evidence of such interactions with other enteric viruses, such as norovirus and poliovirus (Baldrige Science 2015, Kuss Science 2011).

3. EPEC is an attaching/effacing pathogen with a murine counterpart *Citrobacter rodentium*. With the in vivo findings that the increased mucus reduces susceptibility to EPEC infection, the authors should put their findings in context of previous studies looking at the in vivo role of the Muc2 mucin in protection from *C.rodentium* infection (PMID:20485566)

Response: Thank you for this suggestion, we have now included this point of reference on line 170.

Reviewer #3 (Remarks to the Author):

In the manuscript “Astrovirus hijacks goblet cell secretory pathway for replication and alters gut mucus barrier” Cortez et al investigate the tropism of murine astrovirus infection. Cortez et al demonstrate a specific staining pattern of astrovirus infected cells consistent with goblet cells rather than bulk infection of all intestinal epithelial cells. Furthermore, Cortez et al perform single cell sequencing to further define a subset of goblet cells that are infected. Cortez et al interrogate whether mucus production is altered during murine astrovirus infection and the consequence that this has on the microbiome and bacterial infection. The general findings are of great interest; however, the claims of viral hijacking of the secretory pathway are overstated. While this reviewer appreciates the difficulty in interrogating goblet cell function in vivo, the assays performed (DSS and Antibiotic treatment) are too broad and pleiotropic to confidently describe a co-option or hijacking as claimed in the text. Additionally, the scRNA seq and 16S analysis needs to be further expanded to better illustrate the authors points.

Major Points

1. The authors claim in the title and throughout the text (e.g. line 177) that astrovirus co-opts secretory pathway for replication. This statement is too strong given the evidence presented. The authors should consider rewording these statements or provide substantially more evidence to substantiate this claim. These experiments would need to include alterations in mucus production through cytokines (e.g. IL-4) and mouse genetics while quantifying viral infection and goblet cell numbers (which are not done in the DSS or antibiotic experiments currently). Additionally, the authors would also need to provide more direct evidence through an in vitro culturing system such as enteroids that mucus enhanced viral replication.

Response: The Reviewer brings up a fair criticism and we have toned down the title, abstract, and line 110 in the Results in order to better reflect our well-supported conclusion that astrovirus infects actively secreting goblet cells. We now only speculate that this is via the mucus secretory pathway in the Discussion (lines 214-216) based on our EM staining (Figure 1b). In collaboration with another lab we have found that the virus does not replicate in standard enteroid culture (unpublished), but we are refining our methods to ensure we get robust mucus secretion, as we hypothesize that this will be critical to successful propagation (now mentioned in Discussion lines 207, as requested in Minor Point #5 below).

2. The authors should show a co-stain with Muc2 or another Goblet cell marker in Figure 1. Additionally, this should be quantified to demonstrate which percentage of the stained cells are goblet cells.

Response: As mentioned in response to Reviewer #1, co-staining with either Muc2 or alcian blue was incompatible with the ISH procedures, but we have now included staining on serial slides in Figure S1b and

staining at multiple magnifications are included above. We have now quantified the percentage of goblet cells (median=30% of goblet cells/villus, range (0-90%) and included this data on lines 50-51.

3. The authors claim that there are 4 different goblet cell groups but it isn't clear what markers and signatures define these populations. The authors should include further analysis demonstrating which genes for goblet cells are associated with the different subpopulations similar to what has been done for enteroendocrine and tuft cell subpopulations in the paper by Haber et al Nature (PMID: 29144463). This analysis will be useful for future investigations into astrovirus infection of goblet cells.

Response: The work performed by Haber et al. was a comprehensive survey of epithelial cell populations, with targeted validation for subpopulations that were identified. Thus, we do not want to overstate the significance or biological relevance of the subpopulations of goblet cells we identified by bioinformatic analysis. Instead, what our data can best reflect is that 3 of the 4 populations demonstrate a secretory phenotype based on expression of their mucus-related genes (Fig. 2d). However, based on the Reviewer's interest in wanting to further future investigations of astrovirus infection of goblet cells, we have now included a list differentially expressed genes between infected and uninfected cells (Table S1), as was also requested by Reviewer #2.

4. The authors claim that the only 3 of 4 populations are susceptible to MuAstV infection (line 76-77). However, this reviewer questions whether or not the data is robust enough to make this claim based on scRNA sequencing alone. For example, are enough reads in the 1 population of cells sufficient to exclude infection that was missed by the depth limitations of scRNA seq?

Response: We agree with the Reviewer that this is a common limitation of scRNA-seq. However, the goal of this study was to identify the main targets of astrovirus infection. To that end, we endeavored to identify productively infected cells which will contain high viral transcripts and would be easily identified by these methods. We do not expect to have uncovered all cells that are either recently infected or are undergoing abortive infection. Further, the fact that this 1 subpopulation of goblet cells is devoid of viral transcripts is consistent with the ISH staining, which showed that only a subset of goblet cells are infected (lines 50-51).

5. The 16S data needs more analysis to demonstrate the specific changes in the microbiome. For example, the authors should present the relative abundance order/family/genus 16S rDNA sequence assignments in a bar chart (or similar graph) of the major bacterial communities. Additionally, it will be important to note which taxa are discriminate? Is it only the mucus associated bacteria? Does the diversity change with astrovirus infection? These analyses will help put the mucus associated bacteria data in better context.

Response: We have now included the relative abundance in Figure S6a. The mucus-degraders were the only taxa that were discriminate over time and we have now made this clearer on line 155. The overall diversity did not significantly change across days post-infection (now Figure S6b).

6. The authors should note that changes in the microbiome by MuAstV (which may or may not be dependent upon mucus changes) may also contribute to the EPEC infection phenotype.

Response: We agree with the reviewer (and Reviewer #2, Major Point #3) that the reduced neonatal EPEC colonization in astrovirus-infected animals may also be due to changes in the microbiome and we have mentioned this possibility (lines 222-227).

Minor Points

1. The text and figures should be expanded to add clarity from its current short format.

Response: Thank you for the comment. We have added additional clarity to the manuscript.

2. It appears from the scRNA sequencing that only goblet cells are infected by murine astrovirus, but didn't appear to be explicitly stated in the text. Did the authors detect MuAstV reads in other cell populations?

Response: As in response to Reviewer #1 Minor Point #4, we are hesitant to conclude that only goblet cells get infected because it could also infect other cell types but may not be productive as evidenced by Figure S3.

3. Figure 2 gets difficult to read at its current size and resolution. This is particular true of Figure 2D as it is difficult to see the symbols. Perhaps a violin plot may complement this point.

Response: We have expanded this figure and added violin plots (now Figure 2e), which was also requested by Reviewer #1.

4. Text on Figure S4 is very difficult to read and interpret.

Response: We have enlarged the text of this figure for clarity.

5. Does this work have any implications for vitro culturing for MuAstV?

Response: We believe this has significant implications for in vitro culturing, as it would suggest that only culturable methods that fully differentiate epithelial cells to secretory goblet cells will be optimal and accurately reflect infection in vivo (now included in Discussion line 207).

6. The authors should discuss MuAstV tropism in light of the Human Astrovirus tropism in vitro findings by the Wobus lab (PMID: 31671153).

Response: We have added this point of discussion to lines 196-199.

Note: We have also reformatted the manuscript in accordance with the journal guidelines.

REVIEWERS' COMMENTS:

Reviewer #1 (Remarks to the Author):

The authors adequately addressed all major concerns and clarified key ambiguities. No further experiments are suggested. The manuscript will be an important and welcome addition to the field.

I would encourage the authors to include the "negative" IL4 and IL13 data they show in the reviewer response (Reviewer 1 Comment 1). This is useful and important negative data that suggests interactions between enteric viruses, mucus, and the type II immune system are more complicated than previously realized.

Reviewer #2 (Remarks to the Author):

The authors Cortez et al have satisfactorily addressed my major and minor concerns originally outlined in my review. This is an exciting study and I look forward to its publication in Nature Communications.

Reviewer #3 (Remarks to the Author):

The authors have made significant changes to the manuscript and have addressed my concerns.

Please see our responses below in *purple italics*.

REVIEWERS' COMMENTS:

Reviewer #1 (Remarks to the Author):

The authors adequately addressed all major concerns and clarified key ambiguities. No further experiments are suggested. The manuscript will be an important and welcome addition to the field.

I would encourage the authors to include the "negative" IL4 and IL13 data they show in the reviewer response (Reviewer 1 Comment 1). This is useful and important negative data that suggests interactions between enteric viruses, mucus, and the type II immune system are more complicated than previously realized.

The data have now been added to lines 123-137 of the Results and Supplemental Figure 5.

Reviewer #2 (Remarks to the Author):

The authors Cortez et al have satisfactorily addressed my major and minor concerns originally outlined in my review. This is an exciting study and I look forward to its publication in Nature Communications.

Thank you, we appreciate your feedback and support!

Reviewer #3 (Remarks to the Author):

The authors have made significant changes to the manuscript and have addressed my concerns.

Thank you, we appreciate your feedback and support!